# Learning to Generate Diverse Pedestrian Movements from Web Videos with Noisy Labels

**Zhizheng Liu, Joe Lin, Wayne Wu, Bolei Zhou**
Department of Computer Science, University of California, Los Angeles

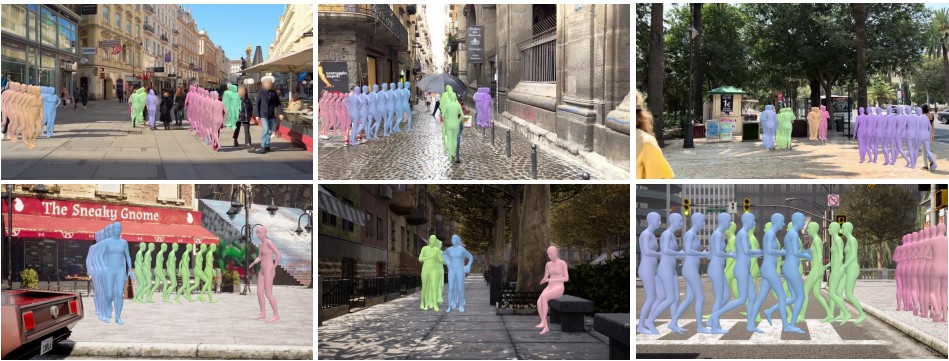

Figure 1: **Pedestrian Movement Generation.** Our method can generate diverse pedestrian movements in real-world (top row) and simulated (bottom row) urban environments.

## Abstract

Understanding and modeling pedestrian movements in the real world is crucial for applications like motion forecasting and scene simulation. Many factors influence pedestrian movements, such as scene context, individual characteristics, and goals, which are often ignored by the existing human generation methods. Web videos contain natural pedestrian behavior and rich motion context, but annotating them with pre-trained predictors leads to noisy labels. In this work, we propose learning diverse pedestrian movements from web videos. We first curate a large-scale dataset called CityWalkers that captures diverse real-world pedestrian movements in urban scenes. Then, based on CityWalkers, we propose a generative model called PedGen for diverse pedestrian movement generation. PedGen introduces automatic label filtering to remove the low-quality labels and a mask embedding to train with partial labels. It also contains a novel context encoder that lifts the 2D scene context to 3D and can incorporate various context factors in generating realistic pedestrian movements in urban scenes. Experiments show that PedGen outperforms existing baseline methods for pedestrian movement generation by learning from noisy labels and incorporating the context factors. In addition, PedGen achieves zero-shot generalization in both real-world and simulated environments. The code, model, and data are available at `https://genforce.github.io/PedGen/`.

## 1 Introduction

In a bustling city, office workers rush through the crosswalk to nearby buildings, tourists wander around along storefronts looking at items on display, and friends sit and enjoy coffee on the outdoor patio. Pedestrians are essential participants in urban spaces. Their movements represent the social lives and interactions with the surrounding environments. Understanding and modeling pedestrian movements is critical to many applications. For example, city designers simulate pedestrian movements to optimize public areas and transportation systems (Mehta, 2008); forecasting the future pedestrian path is crucial for the safe deployment of autonomous vehicles (Lyssenko et al., 2021).

Generating diverse and realistic pedestrian movements remains challenging. Multiple context factors affect pedestrian movements in the scenes. The first factor is the surrounding environment. As

people constantly interact with the environment, it is important to model the scene context where the interaction happens. For example, objects like trash bins, plants, and other pedestrians in the scene can influence the walking behavior of a pedestrian (Daamen & Hoogendoorn, 2003), while the space types like crosswalks and waiting zones decide the overall movement pattern (Sime, 1995). The second factor is the individual characteristics. Studies have shown that walking speed and gait are influenced by age (Ostrosky et al., 1994), gender (Yamasaki et al., 1991), and body weight (Heglund & Taylor, 1988) and can also reflect the pedestrian's fitness level (Dridi, 2015). The last factor is the goal of the pedestrian, which decides the walking route in the scene (Hoogendoorn & Bovy, 2004).

Existing human motion generation research mainly focuses on the breadth of activities (Mahmood et al., 2019), but few have studied generating natural real-world pedestrian movements in diverse scene contexts. For example, existing outdoor human motion datasets have limited scenes and human subjects with unnatural motion performed by actors with specific instructions (Kaufmann et al., 2023; Dai et al., 2023). Moreover, current motion generation methods (Guo et al., 2020; Tevet et al., 2022) lean toward generating complex motions from clean MoCap data, where many motion categories are rare in daily scenes. Few have considered learning the diverse motion contexts from noisy labels.

Web videos, captured by many people walking and touring in different cities worldwide and shared on the YouTube website, contain diverse scene contexts and pedestrian movements in the most natural forms. However, labeling pedestrian movements in these web videos with pretrained predictors leads to inevitable label noise. How to harness the web data with large noise yet rich context becomes the pivot of modeling and generating diverse pedestrian movements. To this end, we first collect *CityWalkers*, a *large-scale* real-world dataset containing pedestrians in urban scenes, annotated with pseudo-labels by off-the-shelf 4D human motion estimation models. CityWalkers captures diverse real-world pedestrian movements regarding various moving speeds, gaits, headings, and local motions. Each movement is also paired with labels of context factors, such as the pedestrian's body shape, route destinations, and the environment's semantics and geometry.

We then develop a new diffusion-based generative model *PedGen* for learning context-aware pedestrian movements with the noisy pseudo-labels from the CityWalkers dataset. PedGen has two key designs: 1) To mitigate the anomaly and incomplete labels from pseudo-labeling techniques, PedGen adopts a data iteration strategy to identify and remove low-quality labels from the dataset automatically and a motion mask embedding to train with partial labels; 2) To model the important context factors, PedGen considers the surrounding environment, the individual characteristics, and the goal points as input conditions to generate realistic and long-term pedestrian movements in urban scenes. As web videos only contain scene context labels in 2D, we propose a novel Context Encoder that can lift the environment context from 2D images into a 3D local scene representation with geometry and semantic information and also encode the other context factors to help generate realistic and long-term 3D pedestrian movements. We show some randomly sampled results of PedGen in Fig. 1.

Experiment results on the CityWalkers validation set, the real-world Waymo open dataset (Sun et al., 2020) and CARLA simulator (Dosovitskiy et al., 2017) show that PedGen can predict more realistic and accurate future pedestrian movements than existing human motion generation methods and achieve better zero-shot generalization by generating high-quality context-aware movements. Additional experiments and ablation studies demonstrate the effectiveness of PedGen in addressing noisy labels and incorporating the key context factors. It enables the application of forecasting pedestrian movements in the real world and populating simulated environments with realistic pedestrians. We summarize *our contributions* as follows: 1) A new task of context-aware pedestrian movement generation from web videos with unique challenges in dealing with label noises and modeling various motion contexts. 2) A new large-scale real-world pedestrian movement dataset CityWalkers with pseudo-labels of diverse pedestrian movements and motion contexts. 3) The context-aware generative model PedGen that can learn from noisy pseudo-labels to generate diverse pedestrian movements.

## 2    RELATED WORK

**Pedestrian Movement Analysis.**    Pedestrian behaviors have been extensively studied in transportation and social science. A hierarchical structure is defined for pedestrian behavior analysis (Hoogendoorn & Bovy, 2004; Feng et al., 2021) from the high-level strategic behavior, the middle-level tactical behavior, to the low-level operational behavior. Pedestrian movements belong to operational behaviors, where pedestrians continuously make short-term movement decisions on their route to respond

to their immediate environment (Daamen, 2002; Duives, 2016). Many works analyze different factors influencing pedestrian movements (Dridi, 2015). Some critical environmental factors include types of space (Sime, 1995), objects in the environment (Daamen & Hoogendoorn, 2003), and movement of other pedestrians (van den Berg, 2016). Pedestrian movements also depend on their biometric data, like age (Ostrosky et al., 1994), gender (Yamasaki et al., 1991), and body size (Heglund & Taylor, 1988). Different from most existing works in pedestrian movement analysis that collect data from field observations (Shields & Boyce, 2000) or controlled experiments (Haghani & Sarvi, 2018) and analyze them using statistical approaches (Tong & Bode, 2023), we extract pedestrian movements from web videos and learn a generative model to facilitate pedestrian movement modeling.

**Human Motion Datasets.** AMASS (Mahmood et al., 2019) is one of the most popular human motion datasets with diverse motions annotated with SMPL (Loper et al., 2023) parameters. The main issue of AMASS is the lack of context information related to the motion. Later datasets have focused on adding more human subjects (Cheng et al., 2023b), text descriptions (Guo et al., 2022), human-object interactions (Bhatnagar et al., 2022) and human-scene interactions (Hassan et al., 2019; 2021; Huang et al., 2022; Jiang et al., 2024) to the labels. Nevertheless, most of these datasets are captured in a controlled environment with the subjects asked to follow specific action instructions, and many motions, such as jump jacks and martial arts, are rarely seen in urban scenes. In-the-wild videos serve as a more suitable source for studying human movement with their richness in individuals and environments. Another line of work (Von Marcard et al., 2018; Guzov et al., 2021; Dai et al., 2023; Kaufmann et al., 2023) focuses on collecting human motion in outdoor places from in-the-wild videos, but their additional sensor requirements like IMUs limit their scalability to collect large-scale datasets. Zhu et al. (2021); Zheng et al. (2022) propose benchmarks for gait recognition in the wild, but these datasets lack diverse scene contexts. Some other datasets (Vendrow et al., 2023; Robicquet et al., 2016) label human social behaviors and trajectories from street videos as key points or bounding boxes. Still, these labels have worse motion granularity than the SMPL parameters for pedestrian movement analysis. Our proposed dataset, CityWalkers, consists of large-scale web videos of pedestrians in diverse urban environments. CityWalkers provides both SMPL movement labels and context pseudo-labels, including the body shape of the pedestrians, their route destinations, and the semantics and geometry of the scene.

**Human Motion Generation.** Human motion generation has been accelerated by large-scale motion datasets and rapid advancements in generative models, with diffusion models (Ho et al., 2020) being the most successful architecture for its high generation quality and multi-modal modeling capacity. To generate motion with more fine-grained control, various input conditions have been used, including action labels (Tevet et al., 2022), texts (Zhang et al., 2024), audios (Dabral et al., 2023), and history motions (Chen et al., 2023a). The most relevant motion generation models to ours are the ones that condition on the indoor scenes (Huang et al., 2023; Yi et al., 2024; Jiang et al., 2024). However, there is a huge gap between indoor and outdoor environments, and generating pedestrian movement requires considering more context factors other than the surrounding environment, like route destinations and pedestrian characteristics. Some works (Tripathi et al., 2024; Xue & Seo, 2024) propose to condition the motion on the body shape but do not consider the scene context. Rempe et al. (2023) animate pedestrian movements by generating high-level trajectories and then training a policy to control the pedestrian movements in simulation. Nevertheless, their movement data for training the RL policy comes from a subset of AMASS (Mahmood et al., 2019) with limited diversity. Wang et al. (2024) can generate diverse pedestrian animations while following the given trajectory, but the motion is not learned from real-world pedestrians. Shan et al. (2023) tackle pedestrian movement generation with rule-based approaches by combining path planning algorithms (Treuille et al., 2006) and manually designed animations, lacking the diversity and realism of real-world pedestrian movements. On the contrary, our proposed PedGen model learns to generate diverse and realistic pedestrian movements conditioned on the context factors from large-scale real-world data with noisy labels.

## 3 CAPTURING DIVERSE REAL-WORLD PEDESTRIAN MOVEMENTS

This section introduces our effort to capture real-world pedestrian movements from web videos. Existing human motion datasets rarely capture natural pedestrian movements, lack diversity in scenes and human subjects, and do not provide the critical context factors of pedestrian movements, such as surrounding environments, individual characteristics, and route destinations. To support the task

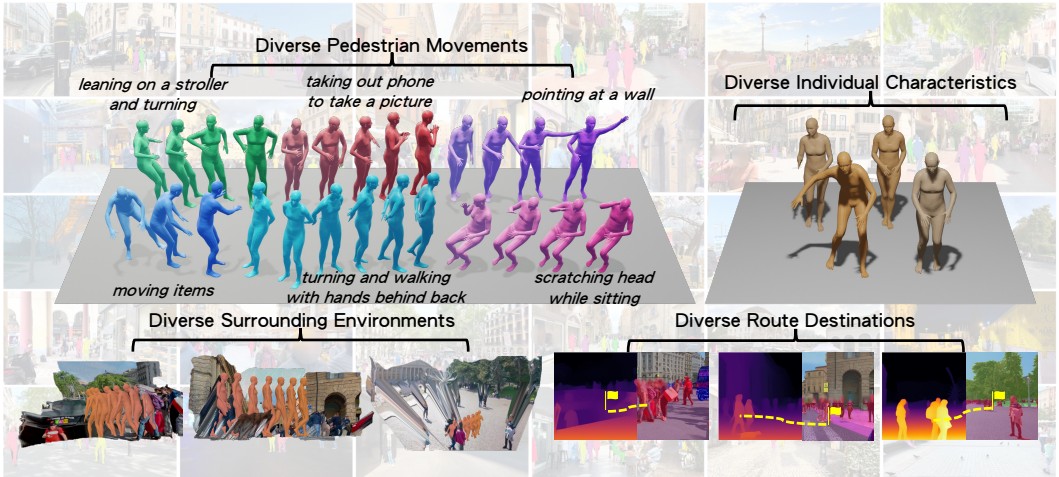

Figure 2: **Samples in the CityWalker dataset.** Top Left: The diverse pedestrian movements. Top Right: The diverse body shapes of the pedestrians. Bottom Left: The diverse surrounding environments from depth-unprojected images and the 4D pedestrian movement labels. Bottom Right: The diverse route destinations shown on the depth labels and semantic maps of the scene. The background showcases pedestrians in bustling cities from where we construct the CityWalker dataset.

of context-aware pedestrian movement generation, we construct CityWalkers, a large-scale dataset with real-world pedestrian movements in diverse urban environments annotated by pseudo-labeling techniques. Sec. 3.1 shows an overview of CityWalkers, and Sec. 3.2 describes our data collection and annotation procedure. Please refer to the appendix for more details about CityWalkers.

## 3.1 CITYWALKERS DATASET

The CityWalkers dataset is collected from YouTube's city tour videos. Our data source consists of high-quality web videos of walking in cities worldwide posted by content creators on YouTube, such as the YouTube channel POPtravel (Sczepansky, 2024). We manually picked videos so they incorporate a wide spectrum of urban public places in different regions and cultures with various scene contexts and real-world pedestrian movements. We label each video's scene attributes, including weather, location, crowd density, and time of the day, using an off-the-shelf VLM (Chen et al., 2023b). We then detect (Jocher et al., 2023) and track (Cheng et al., 2023a) each pedestrian in the video, and label the pedestrian movements as SMPL parameters and scene context as depth and semantic segmentation maps. These labels contain the crucial context factors for learning pedestrian movements. In total, CityWalker contains 30.8 hours of high-quality videos, including 120,914 pedestrians and 16,215 scenes across 227 cities and 49 countries, making it the most diverse human motion dataset regarding scene context and human subjects. As shown in Fig. 2, CityWalkers contains a variety of pedestrian movements, scene contexts, pedestrian body shapes, and route destinations.

## 3.2 DATA PROCESSING AND AUTOLABELING

We adopt WHAM (Shin et al., 2023) to recover 4D pedestrian movement pseudo-labels. Given tracked pedestrians and their body key points, WHAM outputs their global human motion with SMPL (Loper et al., 2023) parameters $\{\boldsymbol{t}_t, \boldsymbol{\phi}_t, \boldsymbol{\theta}_t, \boldsymbol{\beta}\}$ at every timestep $t$. $\boldsymbol{t}_t \in \mathbb{R}^3$, $\boldsymbol{\phi}_t \in SO(3)$ are the root translation and orientation of the SMPL model, and $\boldsymbol{\theta}_t \in SO(3)^{23}$, $\boldsymbol{\beta} \in \mathbb{R}^{10}$ are the body pose and shape parameters. We further use the state-of-the-art monocular depth estimation model ZoeDepth (Bhat et al., 2023) and semantic segmentation model SegFormer (Xie et al., 2021) to automatically label scene depth and semantics. To filter low-quality pedestrian movement labels brought by model noise, we threshold the detection confidence score, human bounding box size, and keypoint confidence scores. We also discard the occluded motion by thresholding the number of human key points that are within the 2D human segmentation mask. To further clean the labels and curate a more accurate evaluation dataset, we manually check the label quality and adjust the labels, such as 2D key points, if necessary. We want to stress that though we have tried our best to improve

the quality of pseudo labels by using state-of-the-art models and filtering wrong predictions, label noise from web videos is still inevitable. We show in the experiments that learning from the noisy labels still benefits pedestrian movement generation. We also apply techniques in our generation model to further mitigate the effect of label noise. A benchmark result of the accuracy of our data autolabeling pipeline is provided in the appendix.

The final label includes future pedestrian movements and their corresponding context factors. $\{x = \{(t_t, \phi_t, \theta_t)\}, y = [\mathcal{I}, \mathcal{I}^d, \mathcal{I}^s, \beta, t_1, t_T]\}$, where $x$ is the pedestrian movement label and $y$ is the context label. $\mathcal{I}, \mathcal{I}^d, \mathcal{I}^s$ are images, depth labels, and segmentation maps of the scene, $\beta$ represents the pedestrian's personal characteristics, and $t_1, t_T$ are the starting and goal positions.

## 4    GENERATING CONTEXT-AWARE PEDESTRIAN MOVEMENTS FROM NOISY LABELS

This section introduces our method, PedGen, to address the label noise and incorporate the context factors from the CityWalkers dataset. PedGen is a diffusion-based generative model and the first method for the new task of context-aware pedestrian movement generation. An overview of PedGen is shown in Fig. 3. Sec. 4.1 defines the task of context-aware pedestrian movement generation. We introduce the overall architecture of PedGen in Sec. 4.2. Sec. 4.3 introduces two simple and effective strategies to deal with the low-quality and incomplete labels from web videos, respectively. As it is crucial to model the context factors, including the surrounding environments, the pedestrian's characteristics, and the goal points during generation, we design a novel Context Encoder in Sec. 4.4.

### 4.1    TASK DEFINITION

We define the task of *context-aware* pedestrian movement generation as follows: Given the initial 3D position $t_1 = [x_1, y_1, z_1]$ of a pedestrian and the 2D image of an urban scene, our goal is to generate the pedestrian's future movements $x = [t_t, \phi_t, \theta_t]_{t=1}^T$. The movement at each timestep is represented as the SMPL root translation $t_t$, root orientation $\phi_t$, and body pose $\theta_t$. The following context factors are also provided to help generation: 1). The urban scene context represented as the 2D image $\mathcal{I}$, semantic mask $\mathcal{I}^s$, and depth label $\mathcal{I}^d$. 2). The SMPL human shape parameter $\beta$, which is a latent representation that can indicate a pedestrian's characteristics, such as height, weight, and body shape. 3). The 3D goal position $t_T = [x_T, y_T, z_T]$ as the pedestrian's route destination in the scene.

### 4.2    PEDGEN MODEL

PedGen follows the conditional diffusion framework (Ho et al., 2020; Ho & Salimans, 2022). Given a sampled movement $x$ from the dataset, the forward diffusion is a Markov noising process $\{x^k\}_{k=0}^K$ defined from the noise scheduling parameter $\{\alpha^k\}_{k=1}^K$. The goal is to train a reverse denoising model $\hat{x} = F(\hat{x}^k, k, c)$ that predicts the sampled movement $x$ given the noisy movement $\hat{x}^k$, the diffusion timestep $k$, and the condition factor $c$. Our denoising model architecture follows state-of-the-art transformer-based human motion generation models (Tevet et al., 2022; Jiang et al., 2024; Chen et al., 2023a), where movements at different timesteps are encoded as separate tokens in the transformer. We represent $x$ as velocity $v_t = t_t - t_{t-1}$ and rotation $\phi_t, \theta_t$ with 6D rotation (Zhou et al., 2019) representation. Different from existing approaches that encode all parts of the motion into a single token, we find it helpful to treat the velocity and the rotation of the motion as different tokens in the transformer, as they have different representations and scales in the data. Our loss function for training the denoising model $F$ is as follows:

$$\mathcal{L}(x, \hat{x}) = \mathbb{E}_{k \in [1,K], (x,c) \in \mathcal{D}}[w_{\text{rec}}\mathcal{L}_{\text{rec}} + w_{\text{traj}}\mathcal{L}_{\text{traj}} + w_{\text{geo}}\mathcal{L}_{\text{geo}}], \qquad (1)$$

where $\mathcal{D}$ is the training dataset. $\mathcal{L}_{\text{rec}} = \|x - \hat{x}\|_2^2$ is the standard diffusion reconstruction loss. $\mathcal{L}_{\text{traj}} = \sum_{t=1}^T \|t_t - \hat{t}_t - t_0\|_1$ is the trajectory loss on the reconstructed global translation $\hat{t}_t = \sum_{t=1}^T \hat{v}_t$, and $\mathcal{L}_{\text{geo}} = \sum_{t=1}^T \|\text{FK}(\theta_t) - \text{FK}(\hat{\theta}_t)\|_2$ is the geometric loss (Tevet et al., 2022) that computes the joint position error with forward kinematics on the SMPL body pose. In ablation experiments, we will show the effectiveness of separating velocity and rotation tokens, as well as adding the trajectory and geometry losses.

During inference, we start from the $k = K$ step by sampling $\hat{\boldsymbol{x}}^K$ from the standard Gaussian distribution. We then pass it to the denoising model to predict the clean motion $\hat{\boldsymbol{x}}$ and add noise again to obtain the noised sample $\hat{\boldsymbol{x}}^{K-1}$. This process is repeated $K$ times to get the final generated pedestrian movement $\hat{\boldsymbol{x}}$. Our model can also facilitate the efficient generation of infinitely long movements by concatenating the short motion intervals. Please refer to the appendix for more details.

## 4.3 Addressing Noisy Labels

Pseudo-labels from pre-trained predictors have inevitable noise. Unlike existing human motion generation approaches that focus on datasets with clean MoCap data, our PedGen model aims to address noisy labels from web videos. There are two sources of label noise in CityWalkers. The first is the inherent compound noise from the data and models used for pseudo-labeling, leading to low-quality labels. As it is difficult to examine the label quality of 4D human motion from only 2D videos, many anomaly labels remain in the dataset even after rule-based filtering and manual checks. We propose to automatically identify these low-quality labels using reconstruction-based unsupervised anomaly detection techniques (Livernoche et al., 2023; Wolleb et al., 2022). Specifically, we first train a PedGen model without context on the training data of CityWalkers. We then partially add noise with half of the diffusion steps $K/2$ for each sample and denoise it using the trained model. We use the reconstruction error between the original and the denoised sample as a metric for anomaly labels and filter labels with errors greater than a certain threshold. We then iterate the above process by re-training the model with the remaining labels and filtering based on the new reconstruction error.

Another source of label noise is the discontinuous or lost tracks of pedestrians due to occlusions and missed detections. As a result, more than half of the labels are incomplete and only annotated on a subset of frames. These labels provide partial supervision that can benefit learning more diverse context-aware pedestrian movements. To train with these partial labels, we replace the missing timesteps for these partial labels with a learnable motion mask embedding $\boldsymbol{m}$ to the denoising transformer. We first define a label mask $\boldsymbol{M}$, where $\boldsymbol{M}_t == 1$ indicates the label at timestep $t$ is missing. Then we add the mask embedding to the original noisy sample as $\boldsymbol{x}^k = \boldsymbol{x}^k(1-\boldsymbol{M}) + \boldsymbol{m} \cdot \boldsymbol{M}$ to replace the missing timesteps with the mask embedding $\boldsymbol{m}$ and feed to the network to output the denoised prediction $\hat{\boldsymbol{x}}$ similar to Sec. 4.2. We then update the loss with the masked predictions and masked ground truth as $L = L(\boldsymbol{x}(\boldsymbol{M}), \hat{\boldsymbol{x}}(\boldsymbol{M}))$ so it only operates on the labels at the available timesteps. All losses will guide the training with the masked outputs.

## 4.4 Context Encoder

Our context encoder module outputs a condition embedding $\boldsymbol{c}$ from the provided context factors. To encode the scene context, we first unproject the 2D depth label $\mathcal{I}^d$ and semantic map $\mathcal{I}^s$ into a 3D point cloud $\mathcal{P} = \{\mathbf{p} = [p_x, p_y, p_z, p_c]\}$, where $p_c$ is the semantic label. Next, we extract points within a local neighborhood of the starting location $\boldsymbol{t}_1$, resulting in the local point cloud $\mathcal{P}_{\text{local}} = \{\mathbf{p} \in \mathcal{P} \mid \|p_x - x_1\| < \Delta_x, \|p_y - y_1\| < \Delta_y, \|p_z - z_1\| < \Delta_z\}$ and voxelize it into a 3D grid. The class label of each voxel is either empty or determined by majority voting of the points within the voxel. The voxel is then processed with a single cross-attention layer to get the scene context embedding. The scene encoding is then added with embeddings from other conditions, including the human shape $\boldsymbol{\beta}$ and the goal position $\boldsymbol{t}_T$, to get the final context embedding $\boldsymbol{c}$.

As web videos only contain 2D images, a natural idea is to directly encode the scene context using the 2D image feature extractors (e.g., Dino-V2 (Oquab et al., 2023)). We find that our proposed context encoder can encode the scene context more effectively than encoding the scene context in 2D. Since pedestrian movement is represented in the 3D space and generating it requires 3D information about the scene, it is easier to reason about the surrounding environment using a 3D representation than 2D image features. Furthermore, it is hard to disentangle the pedestrian and its surrounding context from the 2D image features. As a result, the model may exploit the ego pedestrian instead of the scene context to generate future movements. Our context encoder could address this issue by unprojecting the pixels into 3D point clouds and discarding point clouds belonging to the ego pedestrian. In our ablation experiments, we show the importance of using the 3D scene representation and appending the semantic labels in encoding the scene context.

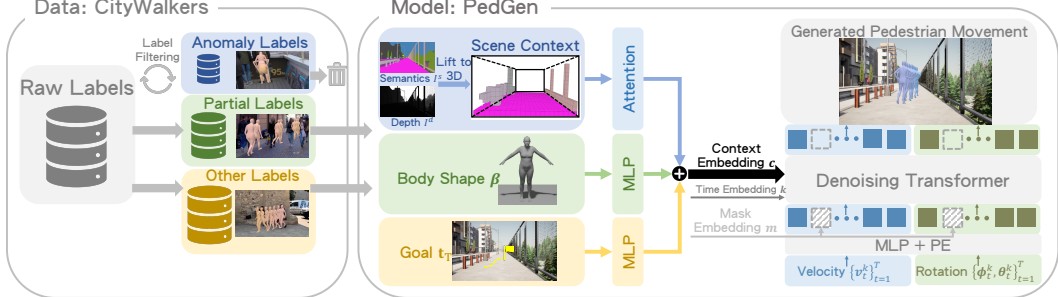

Figure 3: **Our method**. We discard the anomaly labels with an iterative automatic label filtering procedure and add the partial labels to training data. We then train PedGen with a Context Encoder to represent crucial context factors. The scene context is obtained by lifting the 2D depth and semantic labels to the 3D space and converting them into a local voxel representation. The encoded scene context is combined with other context factors, including the body shape and the goal to get the context embedding $c$. The context embedding $c$ and the timestep embedding $k$ are then used to guide the Denoising Transformer to predict the clean motion from the noised one. We use a learnable motion mask embedding $m$ to address the partial labels during training.

## 5 EXPERIMENTS

We compare the performance of PedGen to the other baselines on the real-world Citywalkers and Waymo datasets and in simulated CARLA environments in Sec. 5.1. Sec. 5.2 shows experiments on the effect of training with noisy labels and different context factors. Sec. 5.3 demonstrates the ablation study of our data and model. The diverse pedestrian movements generated by PedGen are shown in Fig. 4. We provide additional experimental details in the appendix.

**Datasets.** We train PedGen on the proposed CityWalkers dataset. For each pedestrian movement trajectory, we sample the initial timestep at an interval of 30 frames and keep at most the future 60 frames (2 seconds) as the ground truth movement. The training set has 104,192 samples, including 53,405 partial labels that have at least 30 frames of annotation. The validation set has 13,039 samples and only contains complete labels. We split the validation set to contain novel scenes with completely different locations and human subjects never seen in the training set.

To verify the performance of PedGen on real-world pedestrian movement prediction with ground truth labels, we use the Waymo open dataset (Sun et al., 2020) with human-annotated 3D human keypoint labels, which is only used for testing. Waymo is critical as it is the largest urban dataset that captures diverse pedestrian motions with sparse 3D keypoint labels at 10 Hz. In total, we selected 80 test samples that (1) spans 2 seconds and (2) includes at least 6 sparse human keypoint labels.

We also collect an additional test set in the CARLA Simulator (Dosovitskiy et al., 2017) to demonstrate the application of PedGen in simulation. We sample initial locations, goal locations, and camera view angles in different maps and render the ground truth images, depth labels, and segmentation maps in simulation. We manually check that all the sampled locations are valid and that all the camera views are not occluded by obstacles, resulting in a simulated test set with 262 diverse samples.

**Evaluation metrics.** For experiments on the CityWalkers and Waymo dataset, we compare the generated pedestrian movement with the ground truth, and follow the metrics used in human motion prediction (Chen et al., 2023a) to compute Average Displacement Error (ADE) and Final Displacement Error (FDE) on the realism of the predicted motion. We generate 50 movements for each data sample and report both minimum and average ADEs (mADE, aADE) and FDEs (mFDE, aFDE) among all samples. For experiments on the simulated test set, we evaluate the context awareness and the physical plausibility of the generated motion. Our metrics include the Collision Rate (CR), which measures the ratio of the generated movement that collides with other objects in the environment, and the Foot Floating Rate (FFR), which measures the ratio of the generated movement whose feet are either floating or penetrating with the ground greater than a given threshold (20cm).

**Baselines.** We compare our method with several recent human motion diffusion models, which have different input conditions and motion representations. Note that since context-aware pedestrian

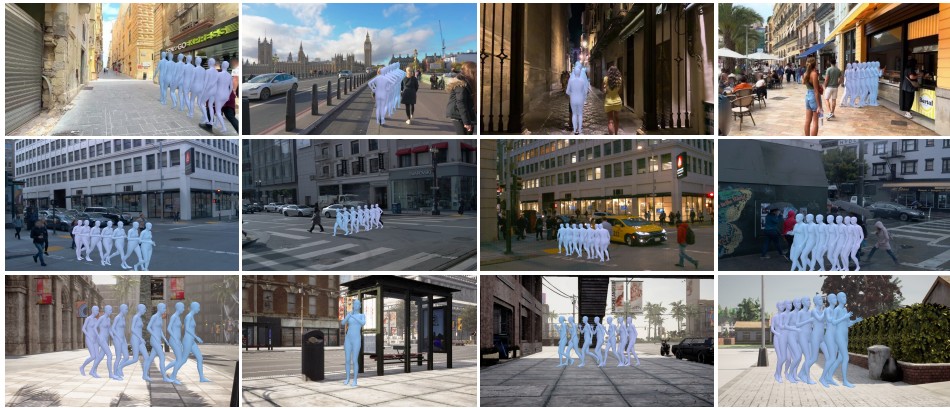

Figure 4: **Visualizations of the generated pedestrian movements.** The top row shows results in real scenes from the CityWalkers dataset, the middle row shows results in the real-world Waymo test set, and the bottom row shows results in simulated scenes from the CARLA test set.

Table 1: **Comparison to baselines.** We consider two cases where the model is given the goal condition or not. We evaluate on the validation set of CityWalkers, the real-world Waymo test set, and the simulated CARLA test set. We evaluate our model (PedGen) compared with the other baselines.

| Goal Condition | Method | CityWalkers | | | | Waymo | | | | CARLA | |
|---|---|---|---|---|---|---|---|---|---|---|---|
| | | mADE ↓ | aADE ↓ | mFDE ↓ | aFDE ↓ | mADE ↓ | aADE ↓ | mFDE ↓ | aFDE ↓ | CR ↓ | FFR ↓ |
| ✗ | HumanMAC | 1.31 | 4.67 | 1.86 | 8.65 | 3.19 | 5.29 | 5.61 | 10.36 | 2.5% | 10.2% |
| ✗ | MDM | 1.33 | 4.55 | 1.93 | 8.41 | 3.03 | 5.35 | 5.66 | 10.60 | 2.1% | 3.2% |
| ✗ | PedGen | **1.13** | **4.08** | **1.61** | **7.56** | **2.90** | **5.15** | **5.52** | **10.11** | **1.6%** | **2.6%** |
| ✓ | TRUMANS | 0.73 | 1.26 | 0.56 | 1.13 | 2.01 | 2.37 | 1.41 | 1.94 | 0.6% | 0.6% |
| ✓ | PedGen | **0.59** | **1.08** | **0.46** | **0.99** | **1.91** | **2.18** | **0.78** | **1.04** | **0.0%** | **0.0%** |

movement generation is a newly defined task, no models could directly support it; we thus made minimum adjustments to make them compatible with our problem setting. MDM (Tevet et al., 2022) uses texts or action labels as the condition and proposes several geometric losses to improve motion quality. We did not use the foot contact loss proposed in their method as CityWalkers does not provide ground-truth foot contact labels. HumanMac (Chen et al., 2023a) conditions the future motion generation on the history motion sequences, which uses the DCT transform to ensure smooth motion generation. TRUMANS (Jiang et al., 2024) uses the initial human pose, the 2D BEV goal position, the indoor scene context, and the frame-wise action labels as the input condition. We modify TRUMANS so it is compatible with our scene context and goal point conditions.

## 5.1 COMPARISON TO BASELINES

We compare PedGen with the other baselines on the validation set of CityWalkers, the real-world Waymo test set, and the simulated CARLA test set. We separately evaluate whether the goal condition is given, as it is a deterministic factor for pedestrian movement. The benchmarking results are shown in Tab. 1. It can be observed that PedGen outperforms other baseline models by a clear margin on CityWalkers. PedGen also achieves the best zero-shot generalization ability on Waymo and CARLA test sets, further showing the superiority of PedGen to the other baseline.

## 5.2 EFFECT OF NOISY LABELS AND CONTEXT FACTORS

Table 2a demonstrates the effectiveness of PedGen in addressing noisy labels. We can see that removing the anomaly labels with the proposed automatic label filtering can help generate more realistic pedestrian movements and improve aADE by 2.9%. Moreover, adding the partial labels as additional training data can improve the aADE by 5.8%, highlighting the value of partial labels in web videos. Combining both strategies for noisy labels leads to the best performance of a 4.08 aADE.

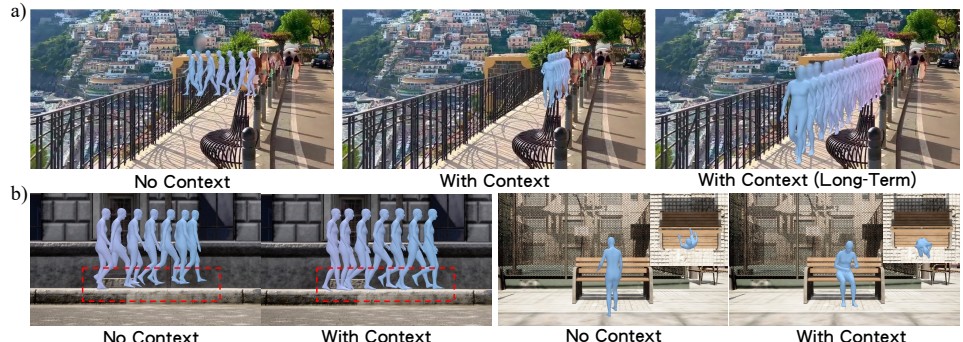

Figure 5: **Qualitative comparison of training with context factors.** We compare the generated movements of PedGen trained with or without context factors in real-world environments (a) and in simulation (b).

Table 2: **Results with noisy labels and context factors.** We experiment on the CityWalkers validation set and the CARLA test set and study the effect of training with noisy labels (a) and different context factors (b).

(a) **Evaluation of training with noisy labels.** We evaluate PedGen with no context trained with or without anomaly and partial labels.

| Anomaly Labels | Partial Labels | CityWalkers | | | |
|---|---|---|---|---|---|
| | | mADE ↓ | aADE ↓ | mFDE ↓ | aFDE ↓ |
| ✓ | ✗ | 1.17 | 4.45 | 1.64 | 8.31 |
| ✗ | ✗ | 1.13 | 4.32 | **1.60** | 8.09 |
| ✓ | ✓ | 1.15 | 4.19 | 1.62 | 7.79 |
| ✗ | ✓ | **1.13** | **4.08** | 1.61 | **7.56** |

(b) **Evaluation of the context factors.** We evaluate PedGen conditioned on each context factor, including the surrounding environment (scene), the pedestrian's own characteristics (human), and the goal points (goal).

| Idx | Context Factor | CityWalkers | | | | CARLA | |
|---|---|---|---|---|---|---|---|
| | | mADE ↓ | aADE ↓ | mFDE ↓ | aFDE ↓ | CR ↓ | FFR ↓ |
| 1 | No | 1.13 | 4.08 | 1.61 | 7.56 | 2.1% | 5.2% |
| 2 | Scene | 1.11 | 3.75 | 1.55 | 6.92 | 1.6% | 2.6% |
| 3 | Human | 1.09 | 3.24 | 1.61 | 5.95 | 1.9% | 0.7% |
| 4 | Scene + Human | **1.00** | **3.12** | **1.43** | **5.65** | **1.5%** | **0.3%** |
| 5 | Goal | 0.60 | 1.09 | 0.47 | 1.00 | 0.0% | 0.0% |
| 6 | Scene + Goal | 0.59 | 1.08 | 0.46 | 0.99 | 0.0% | 0.0% |
| 7 | Human + Goal | 0.55 | 1.01 | 0.43 | 0.93 | 0.0% | 0.0% |
| 8 | Scene + Human + Goal | **0.54** | **0.96** | **0.43** | **0.91** | **0.0%** | **0.0%** |

We further examine the effect of each context factor on the pedestrian movement generation performance in Tab. 2b. The comparisons between "setting 1 vs. 2" and between "setting 1 vs. 3" show that adding the surrounding environment and the pedestrian's own characteristics as conditions are helpful to pedestrian movement generation and can reduce the aADE by $6.7\%$ and $19.4\%$, respectively. Setting 4 demonstrates that incorporating both context factors achieves even better results. From settings 1 and 5, we find that the goal points are the most important context factor for the final performance and can significantly reduce aADE by $72.9\%$. The comparison between "setting 5 vs. 6" and between "setting 5 vs. 7" further proves that the other two context factors (scene and human) can also help movement generation after the goal is provided. Finally, setting 8 demonstrates that using all three context factors leads to the smallest generation errors. From the results of the CARLA test set, we can see that the scene context can contribute to both a lower collision rate of 1.6% and a lower foot floating rate of 2.6%, while the human context is more useful in reducing the foot floating rate to only 0.7%. Adding both the scene and the human context can further improve the physical plausibility of the generated movements. Using the goal context is the most crucial factor and can reduce the failure rate to 0. This further proves the effectiveness of each context factor.

Some qualitative comparison results in real-world environments are shown in Fig. 5 (a). We can observe that PedGen trained without context factors generate arbitrary movements that walk off the sidewalk. By conditioning the generation on the context factors, the movement becomes context-aware, and the model can further generate long-term pedestrian walking behaviors on the sidewalk. Visualization results in simulation are shown in Fig. 5 (b). From the left two figures, we can see that without the context factors, the generated movements have the feet floating from the ground, while adding context factors fix this issue. From the right two figures, we see that PedGen trained without context generated a walking pose that is in collision with the bench, while it successfully generated a sitting pose on the bench after considering the context factors.

Table 3: **Ablation experiment results.** We ablate on the training data of PedGen (a) and the PedGen model's key components (b) on the CityWalkers validation set.

(a) Ablation on training data of PedGen.

| Context Factor | Training Data | Metric | | | |
|---|---|---|---|---|---|
| | | mADE ↓ | aADE ↓ | mFDE ↓ | aFDE ↓ |
| No | SLOPER4D | 1.61 | 6.04 | 2.42 | 11.65 |
| | CityWalkers(50%) | 1.16 | 4.19 | 1.66 | 7.76 |
| | CityWalkers(100%) | **1.13** | **4.08** | **1.61** | **7.56** |
| Scene | SLOPER4D | 1.47 | 6.45 | 2.19 | 12.50 |
| | CityWalkers(50%) | 1.14 | 4.02 | 1.57 | 7.47 |
| | CityWalkers(100%) | **1.11** | **3.75** | **1.55** | **6.92** |
| Human | SLOPER4D | 3.82 | 10.39 | 7.11 | 20.62 |
| | CityWalkers(50%) | 1.13 | 3.34 | 1.65 | 6.18 |
| | CityWalkers(100%) | **1.09** | **3.24** | **1.61** | **5.95** |

(b) Ablation on model components of PedGen.

| Context Factor | Method | Metric | | | |
|---|---|---|---|---|---|
| | | mADE ↓ | aADE ↓ | mFDE ↓ | aFDE ↓ |
| No | PedGen (-traj/geo) | 1.33 | 4.47 | 1.97 | 8.29 |
| | PedGen (-sep. token) | 1.32 | 4.26 | 2.01 | 7.95 |
| | PedGen | **1.13** | **4.08** | **1.61** | **7.56** |
| Scene | PedGen (-3D rep.) | 1.13 | 4.26 | 1.60 | 7.95 |
| | PedGen (-semantic) | 1.12 | 3.86 | 1.56 | 7.09 |
| | PedGen | **1.11** | **3.75** | **1.55** | **6.92** |

## 5.3 ABLATION STUDY

**Ablation on training data of PedGen.** Table 3a shows an ablation study on the training data of PedGen. To demonstrate the effectiveness and necessity of using web videos for pedestrian movement generation, we train PedGen on SLOPER4D (Dai et al., 2023). SLOPER4D is one of the largest outdoor 4D human datasets annotated from LiDAR point clouds. Still, its scale and diversity in human subjects and scene contexts are much less than CityWalkers, and its motion is captured from human actors instead of real-world pedestrians. We can observe that training on CityWalkers leads to significant performance improvement compared to training on SLOPER4D, even though CityWalkers is annotated with pseudo-labels. Notably, training with the human context on SLOPER4D only achieves 3.82 mADE, as the model can easily overfit the 12 human subjects in the dataset and results in degraded generalization performance. On the contrary, with more training data of CityWalkers, models conditioned on different context factors all perform better than those trained with less data, highlighting the value of capturing large-scale pedestrian movement data using web videos.

**Ablation on model components of PedGen.** Table 3b shows ablations on key model components of PedGen. *-traj/geo* means the model is trained only with the diffusion reconstruction loss without the trajectory and geometry losses. *-sep. token* means the motion at each timestep is represented as a single token in the transformer instead of separate tokens for the rotation and velocity. *-3D rep.* means the scene context is encoded with 2D feature using a depth anything (Yang et al., 2024) pre-trained image backbone DINOv2 (Oquab et al., 2023). *-semantic* means the 3D voxel only encodes the occupancy without semantic labels. The results show the importance of using 3D scene context with semantic labels for pedestrian movement generation, the effectiveness of separating velocity and rotation tokens, and adding the geometry and trajectory losses in the motion diffusion model.

## 6 CONCLUSION

We study a new task of generating context-aware pedestrian movements by learning from web videos with noisy labels. To facilitate this study, we collect a large-scale dataset CityWalkers with diverse real-world pedestrian movements in urban scenes. We further propose PedGen, a generative model that addresses the noisy labels in CityWalkers and models the three important context factors: the scene context, the personal characteristics, and the goal position. Experiments show that PedGen can generate realistic context-aware pedestrian movements in both real-world and simulation environments. We hope this study will present new opportunities and facilitate future research on modeling pedestrian movements in real-world settings.

**Limitations.** PedGen only considers static scene context at the starting frame, while pedestrian movement also depends on dynamic scene contexts such as the history trajectories of other pedestrians. Modeling these dynamic objects would be an interesting future direction. In addition, PedGen only generates the movements of a single pedestrian at one time, while modeling group activities in urban scenes can help generate more realistic real-world behaviors.

ETHICS STATEMENT

We are fully aware of the potential privacy and ethical issues when using street-view videos, and we take these concerns as our highest priority. We acknowledge that we will be solely responsible for any legal violations with respect to our collected dataset and that we accept all the associated risks. We have taken several measures to mitigate these issues:

**Legal Compliance with YouTube's Terms of Service and Creative Commons Licensing**: Our dataset comprises YouTube videos that are explicitly labeled under the Creative Commons Attribution (CC-BY-SA[1]) license. This license allows for redistribution and adaptation of the material for any purpose, including research, as long as proper attribution is provided to the original creators. We have carefully reviewed YouTube's Terms of Service and Creative Commons guidelines [2] to ensure that our use of these videos is legally permissible. YouTube's Terms of Service explicitly permit redistribution of Creative Commons-licensed content, provided proper attribution is given. Our dataset strictly excludes videos marked under the "Standard YouTube License" or any content that carries additional restrictions. We do not download videos through unauthorized means but instead rely on publicly available Creative Commons content accessible via YouTube's official API, ensuring compliance with both YouTube's platform policies and broader copyright laws.

**Privacy Protection and Anonymization Measures**: We follow standard protocols used by street datasets such as Waymo and Ego4D (Sun et al., 2020; Grauman et al., 2022) to ensure complete anonymization of identifiable features. As the first step of our data preprocessing, we apply mosaics to faces and license plates to prevent re-identification of individuals in the videos. Additionally, we omit any copyright-related information, such as logos, channel owner details, or other copyrighted materials, during data processing to protect intellectual property rights. Our model is trained exclusively on preprocessed images, ensuring that identifiable features and copyrighted materials are neither stored nor accessible within our dataset. Furthermore, our IRB-approved protocol ensures compliance with human subjects protection guidelines, giving individuals in the videos the right to request data removal if privacy concerns arise.

**Ethical Use and Responsible Data Sharing**: Upon data release, we will provide clear and transparent terms of use for the dataset, outlining ethical guidelines, usage restrictions, and legal obligations that users must comply with. We will implement safeguards on our dataset webpage, including detailed user agreements, encryption measures to protect personal information, and access controls to monitor and prevent unauthorized data use. To align with privacy protection best practices from existing YouTube datasets such as YouTube-8M and YouTube-VOS (Abu-El-Haija et al., 2016; Xu et al., 2018), we will not provide processed video clips directly. Instead, users will be redirected to the original YouTube videos via a link and will be required to follow our preprocessing scripts to prepare the dataset themselves. Additionally, as our labels are generated using off-the-shelf models, we will not release direct annotations but will provide scripts that allow users to generate their own pseudo-labels. To further prevent misuse, we will only release the trained model weights, allowing researchers to utilize the dataset for downstream tasks such as pedestrian movement simulation without exposing raw video data.

**Institutional Review Board (IRB) Oversight and Legal Compliance**: Our dataset will be overseen by the UCLA Institutional Review Board (IRB) with security protocols in place to protect against unauthorized access and misuse. The UCLA IRB will review our data collection, ensuring compliance with ethical and legal standards. Specifically, the IRB will assess the legal permissibility of using publicly available, CC-licensed videos and will confirm that our approach aligns with applicable laws and institutional policies. The human subjects in the videos have the right to view, correct, and request deletion of personal information in the dataset. The review board will continue to evaluate the collection, use, and sharing of the data to ensure alignment with best practices in data privacy and ethics. Any legal concerns raised post-publication will be addressed under UCLA's data governance framework, ensuring continued compliance and ethical responsibility.

By implementing these measures, we ensure that our dataset adheres to both legal requirements and ethical best practices. We remain committed to continuously reviewing and improving our compliance protocols in response to emerging ethical and legal considerations.

---

[1] https://creativecommons.org/licenses/by/3.0/legalcode.en
[2] https://www.youtube.com/static?template=terms

ACKNOWLEDGMENTS

The project was supported by the NSF grants CNS-2235012 and IIS-2339769.

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
