# OpenReview forum: "Learning to Generate Diverse Pedestrian Movements from Web Videos with Noisy Labels"
_ICLR.cc/2025/Conference — ICLR 2025 Poster_

### Official Review · Reviewer_XLS3 · 2024-10-16

**Soundness:** 4
**Presentation:** 4
**Contribution:** 3
**Rating:** 8
**Confidence:** 3

**Summary:**

This paper studies a new task of generating context-aware pedestrian movements by learning from web videos with noisy labels. Different from previous human motion generation task that focuses on in-door human motion generation, the proposed method can generate outdoor human movements conditioned on scene information and person identity information. To achieve this goal, this paper also introduces a new dataset named as CityWalker, which contains outdoor urban pedestrian movements from web videos. Experiments on several benchmarks demonstrate the effectiveness of the proposed method.

**Strengths:**

1. The contribution of this paper is comprehensive because it contains both dataset and method, which can provide reference for future research in this area.
2. The focus of this paper is interesting, generating human motion in outdoor environment is not well-studied by previous work and this paper present a solution to this task, which may be useful for autonomous driving and related field.
3. This paper is well-written and easy to understand.

**Weaknesses:**

Although this paper clarifies their task as motion generation, the proposed method and evaluation metrics are more like motion prediction, i.e., output person movements by accepting current condition inputs. The evaluation metrics also does not involve the diversity evaluation of the generated human motion. Therefore I have doubts about the task type of this paper, maybe motion prediction is more accurate.

**Questions:**

In weakness.

---

> ### Author Response · Authors · 2024-11-20
>
> Thank you for your valuable and helpful feedback. We sincerely appreciate the supportive comments that "The contribution of this paper is comprehensive," “The focus of this paper is interesting,” and “This paper is well-written and easy to understand.” We address your question below.
>
> **Q1**. Although this paper clarifies their task as motion generation, the proposed method and evaluation metrics are more like motion prediction, i.e., output person movements by accepting current condition inputs. The evaluation metrics also does not involve the diversity evaluation of the generated human motion. Therefore I have doubts about the task type of this paper, maybe motion prediction is more accurate.
>
> **A1**. In our setting, defining our task as pedestrian movement generation is more relevant to the key motivation of PedGen, which is populating empty urban spaces in simulation by generating realistic and diverse pedestrian movements (shown in the supp. video). We also evaluate the context awareness and the physical plausibility of the generated movements in simulated environments on the Carla test for this application. Another application is to predict future pedestrian movements in the real world, and hence we use motion prediction metrics like ADE and FDE. For diversity evaluation, we use average pairwise distance (APD) between all predictions and evaluate the result on each context factor in the table below:
> | Context Factor | APD  |
> |----------------|------|
> | No             | 38.6 |
> | Scene          | 29.8 |
> | Human          | 21.7 |
> | Goal           | 10.2 |
>
>
> The results show that the model with no context achieves the best diversity of 38.6 APD, whereas adding the goal context reduces APD to only 10.2. This is expected as adding context factors reduces diversity by eliminating the implausible results. Therefore, we do not add APD as our metric, since comparing the generation diversity of models with different context factors is meaningless in our setting.

---

> > ### Author Response · Authors · 2024-11-27
> >
> > Dear Reviewer XLS3,
> >
> > Thanks again for your supportive feedback!  We would appreciate it if you would let us know whether our responses sufficiently address your questions and whether you need to see additional visualizations, as the last day to update the materials is Nov. 27th. Thank you once again!
> >
> > Best regards, Authors of Submission ID 1930

---

### Official Review · Reviewer_yQtW · 2024-10-23

**Soundness:** 3
**Presentation:** 2
**Contribution:** 2
**Rating:** 5
**Confidence:** 4

**Summary:**

In this work, the authors propose to learn pedestrian movements from web videos. To this end, they curate a large-scale dataset called CityWalkers that captures real-world pedestrian movements in urban scenes from YouTube, with some extra efforts to generate pseudo-GT labels and remove the low-quality labels. Given this dataset, the authors propose a generative model where a context encoder is introduced to incorporate various context factors, including goal, human, and scene, in generating realistic pedestrian movements in urban scenes.

Experiments show that the proposed outperforms existing baseline methods for pedestrian movement generation by learning extra data and incorporating the context factors.

**Strengths:**

1. The concept of integrating contextual information into pedestrian generation is sound and has already demonstrated effectiveness in related trajectory generation tasks.
2. The ablation studies examining each factor are thorough and well-executed.
3. The visual examples provided in both the main paper and supplementary materials are satisfactory.

**Weaknesses:**

1. Unclear Task Definition: Section 4.1 outlines the overall task definition, but it seems that only the 3D location of pedestrians and the 2D image of the scene in the first frame (t_1) are provided, while all other elements are predicted. Is this interpretation correct? If so, what is actually learned from this setup, considering there could be millions of plausible results? Or do we learn the bias in the training set? Is this a reasonable setup?
2. Clarification of Contributions: The overall contributions should be clarified. Lines 97-101 describe the contributions in a confusing manner. For instance, line 98 states, “1) A new task of context-aware pedestrian movement generation from web videos with unique challenges in dealing with label noise and modeling various motion contexts.” This seems contradictory since the dataset with noisy labels is presented as a contribution in “A new large-scale real-world pedestrian movement dataset, CityWalkers, with pseudo-labels of diverse pedestrian movements and motion contexts” (l. 99). I have significant doubts about the dataset's quality and question whether a noisy dataset can genuinely be considered a contribution.
3. Need for More Explanations About Results: Based on Table 2(b), incorporating scene information yields only minor improvements. The priority order appears to be Goal, Human, and Scene, which raises questions about the usefulness and necessity of including Scene in the overall model. The authors should provide an explanation for this observation.
4. Visual Results for Complex Scenes: More visual examples are needed to illustrate how pedestrians navigate around obstacles, such as chairs, trees, or cars, to reach their targets. Without these examples, the scene context seems to have limited utility, as indicated by the table.

**Questions:**

Please address my concerns in weakness as well as in below.

1. Given that the ground truth (GT) is primarily generated using existing methods, how do the authors ensure consistency across these various methods? For example, is the generated depth map aligned with the generated 4D human pose? If there is a discrepancy, what potential drawbacks arise from this mismatch, and how can the proposed method address them?
2. How do the authors convert the depth labels into a 3D point cloud without knowing the camera parameters (l.310)? Am I missing any assumptions here?
3. It appears that only static objects and elements are considered in the scene context, as there is no explicit modeling of dynamics in the context encoder. Do the authors intentionally exclude dynamics during scene context modeling, or are these dynamics treated as static objects? Is it valid to assume that other road participants, such as pedestrians, are not significant in the modeling process for pedestrian motion generation? I find this assumption questionable, especially since collisions are used as evaluation metrics in this study.

**Details Of Ethics Concerns:**

The authors address potential ethical issues in both the main paper and supplementary materials.

As I am not an ethics reviewer, I would like to highlight these concerns and recommend that they be reviewed by experts in the field.

---

> ### Author Response · Authors · 2024-11-20
>
> Thank you for your thorough and insightful comments. We sincerely appreciate the supportive feedback that  “the concept of integrating contextual information into pedestrian generation is sound," "the ablation studies examining each factor are thorough and well-executed”, and "the visual examples provided in both the main paper and supplementary materials are satisfactory." We address your questions below.
>
> **Q1**. Unclear Task Definition: Section 4.1 outlines the overall task definition, but it seems that only the 3D location of pedestrians and the 2D image of the scene in the first frame (t_1) are provided, while all other elements are predicted. Is this interpretation correct? If so, what is actually learned from this setup, considering there could be millions of plausible results? Or do we learn the bias in the training set? Is this a reasonable setup?
>
> **A1**. Yes, the model is only given the starting 3D location of the pedestrian and the context factors in the first frame. As a generative model, we aim to capture the real-world pedestrian movement ***distribution*** conditioned on the context factors. We believe such a setting is reasonable as we can match the real-world distribution by leveraging large-scale training data and diffusion models so the generated movements are natural and diverse. While the model can generate millions of plausible results, it learns the posterior distribution of which movements are more likely given the current context. Zero-shot generation experiments on the Waymo dataset and the Carla test set in Tab.1 further validate our model’s strong generalization ability, where it improves mADE by ***0.13*** on Waymo and collision rate by ***0.5%*** on CARLA, in comparison to the best-performing baselines. Thus, we can leverage the learned real-world distribution to populate more realistic pedestrian movements in simulation environments instead of fixed animations.
>
>
> **Q2**. Clarification of Contributions: The overall contributions should be clarified. Lines 97-101 describe the contributions in a confusing manner. For instance, line 98 states, “1) A new task of context-aware pedestrian movement generation from web videos with unique challenges in dealing with label noise and modeling various motion contexts.” This seems contradictory since the dataset with noisy labels is presented as a contribution in “A new large-scale real-world pedestrian movement dataset, CityWalkers, with pseudo-labels of diverse pedestrian movements and motion contexts” (l. 99). I have significant doubts about the dataset's quality and question whether a noisy dataset can genuinely be considered a contribution.
>
> **A2**. The two points of our contribution are not contradictory but complementary.  Our CityWalkers dataset is a large-scale dataset that has the most diverse pedestrian movements and motion contexts compared to existing human motion datasets. Our PedGen model aims to learn from the diverse labels and harness the inherent noise from large-scale data by leveraging the partial labels and filtering out the noisy labels. Ablation studies in Tab.3a further demonstrate that training with CityWalkers with noisy labels significantly improves performance than training on a smaller-scale dataset SLOPER4D with ground truth labels, where training with CityWalkers achieves a mADE of ***1.09*** using the human context compared to a mADE of ***3.82*** from training with SLOPER4D. This could show the utility of CityWalkers in capturing large-scale real-world pedestrian movements, albeit inherent noises.

---

> ### Author Response · Authors · 2024-11-20
>
> **Q3**. Need for More Explanations About Results: Based on Table 2(b), incorporating scene information yields only minor improvements. The priority order appears to be Goal, Human, and Scene, which raises questions about the usefulness and necessity of including Scene in the overall model. The authors should provide an explanation for this observation.
>
> **A3**: It is necessary to include the scene context in the PedGen model, as the experiment results show all three context factors can lead to improvements on top of each other, and incorporating all context factors lead to the best performance. Also, the performance gain from incorporating the scene context is already significant. As shown in Tab.2, filtering out the noisy labels and adding the partial labels improves mFDE from ***1.64*** to ***1.61***, while using the scene context further improves mFDE to ***1.55***. It is worth noting that we use motion prediction metrics ADE and FDE in our real-world experiments by comparing the error between the generated movements and the ground truth. The results can not truly showcase the effectiveness of the scene context, as the ground truth is only one of the plausible results, and the scene context is more useful in eliminating bad predictions that collide with other objects rather than making the predictions match exactly to the ground truth. To better showcase the effectiveness of the scene context in other metrics, we have conducted additional ablations of the context factors in the CARLA simulator and evaluated the performance using collision rate and foot floating rate. The results are shown below:
>
> | Context Factor | Collision Rate % | Foot Floating Rate % |
> |----------------|------------------|----------------------|
> | No             | 2.1              | 5.2                  |
> | Scene          | 1.6              | 2.6                  |
> | Human          | 1.9              | 0.7                  |
> | Scene+Human    | 1.5              | 0.3                  |
> | Goal           | 0.0              | 0.0                  |
>
> We can see that the scene context is more helpful than the human context in reducing the collision rate (***0.5%*** improvement compared to ***0.2%***), while the goal remains the most critical context factor. We will add this experiment to our updated paper.
>
> **Q4**. Visual Results for Complex Scenes: More visual examples are needed to illustrate how pedestrians navigate around obstacles, such as chairs, trees, or cars, to reach their targets. Without these examples, the scene context seems to have limited utility, as indicated by the table.
>
> **A4**: We will include more visual results in the supplementary to demonstrate the effectiveness of the scene context in obstacle avoidance.
>
>
> **Q5**. Given that the ground truth (GT) is primarily generated using existing methods, how do the authors ensure consistency across these various methods? For example, is the generated depth map aligned with the generated 4D human pose? If there is a discrepancy, what potential drawbacks arise from this mismatch, and how can the proposed method address them?
>
> **A5**:  We have mentioned in supp. Sec. C the potential inconsistencies between the depth map label and the 4D human pose label and outline our ways to address this issue. To summarize, we multiply the depth map label by a factor $\gamma$, which equals the ratio between the depth from the SMPL root translation of the first frame and the depth of the human root’s projection in the 2D depth map label to align the starting position of the motion and its surrounding scene context. As shown in the bottom left examples of Fig.2, the scene and movement labels fit well after alignment.
>
> **Q6**. How do the authors convert the depth labels into a 3D point cloud without knowing the camera parameters (l.310)? Am I missing any assumptions here?
>
> **A6**.  As stated in supp. Sec. C, we estimate the camera intrinsics by setting the focal length to be the diagonal pixel length of the image and the optical center to be the center of the image. While such estimation may lead to additional errors, our experiments in Tab. 2b show that learning from the noisy scene context labels can still benefit pedestrian movement generation compared to using no context, reducing aADE and aFDE by ***6.7%*** and ***8.5%***, respectively.

---

> ### Author Response · Authors · 2024-11-20
>
> **Q7**. It appears that only static objects and elements are considered in the scene context, as there is no explicit modeling of dynamics in the context encoder. Do the authors intentionally exclude dynamics during scene context modeling, or are these dynamics treated as static objects? Is it valid to assume that other road participants, such as pedestrians, are not significant in the modeling process for pedestrian motion generation? I find this assumption questionable, especially since collisions are used as evaluation metrics in this study.
>
> **A7**: As discussed in our global response, the key motivation of our model is to **populate empty urban spaces in simulation by generating realistic and diverse pedestrian movements**, as shown in the supp. video. Therefore, we only consider the static environment so we can start generation from an empty static scene without pedestrians, though it is possible to extend our model to incorporate historical information. In addition, as our focus is more on generating detailed human body movements as SMPL meshes than the global trajectory, we assume a path planner like A* already models the scene dynamics when predicting the global path. We could further iteratively generate future movements based on the context at the latest time step so the model can adaptively update its predicted movements according to the behaviors of other pedestrians. In training, when extracting the scene context in urban environments, the context often includes road participants like other pedestrians and local obstacles such as benches and fences, so the model has already learned some collision avoidance capabilities implicitly (as shown in the ablation table above, adding the scene context reduces collision rate from ***2.1%*** to ***1.6%***).

---

> > ### Comment · Reviewer_yQtW · 2024-11-25
> >
> > Apologies for the confusion. I did watch the supplementary videos and noticed that the authors included scenarios set in realistic environments with other pedestrians and vehicles present. These videos are quite misleading and raise questions about how the authors account for these potentially dynamic environments. Could the authors elaborate on how these videos were generated or provide more details on how such dynamics are addressed?
> >
> > Additionally, while the approach to "populate empty urban spaces in simulation by generating realistic and diverse pedestrian movements" is certainly valid, I would argue that this setup is somewhat limited, particularly since the environment has already been factored in. More importantly, regarding the claim that "While the model can generate millions of plausible results, it learns the posterior distribution of which movements are more likely given the current context," I have questions about the definition of "current context." From the dataset videos, it appears the "current context" includes both static elements (e.g., the environment) and dynamic elements (e.g., surrounding or accompanying pedestrians). How do the authors ensure the model learns appropriately from a "context" that integrates both static and dynamic factors, while aiming to make predictions under static scenarios alone? In other words, how do the authors decouple the static and dynamic components in the learning process to achieve this separation effectively?

---

> ### Author Response · Authors · 2024-11-25
>
> Thanks for the further discussion. We will clarify the first point. We believe that modeling the real-world pedestrian movement distribution $Q(X)$  is complicated, and many more factors can influence the posterior distribution of pedestrian movement in addition to the dynamic scene context, such as the weather, social norms, group dynamics, and a person’s mood.  As the first paper addresses the task of pedestrian movement generation, it is challenging to incorporate and validate all the possible context factors and learn the exact real-world distribution $Q(X) = P(X|Y_1, Y_2, ..., Y_N)$. We believe deciding which context factor $Y_i$ should be included in the model depends on its application. For example, in trajectory prediction in autonomous driving, $Y_i$ can include the dynamics of other agents, but it does not contain personal characteristics as it does not require predicting the local body movements. However, as the main motivation of PedGen is to populate empty urban spaces in simulation with detailed body movements, we identify the three most fundamental factors for our application to be $Y_1$=static scene context, $Y_2$=personal characteristics, and $Y_3$=goal points. These three factors are already sufficient to support our main task, and hence, we only aim to learn the distribution $P(X|Y_1=(\mathrm{scene}), Y_2=(\mathrm{human}), Y_3=(\mathrm{goal}))$. As for the scene context, our goal is not to show the real-world distribution $Q(X)$  is equivalent to $P(X|Y_1=(\mathrm{scene}))$ but to prove the effectiveness of each factor by showing $Q(X)$  is closer to   $P(X|Y_1=(\mathrm{scene}))$ than $P(X)$. As CityWalkers is a large-scale dataset, the learned posterior distribution $P(X|Y_1=\mathrm{scene})$ has strong generalization ability. As shown in Tab. 2b, incorporating the scene context can reduce aADE from 4.08 to 3.75 on the validation set of CityWalkers with novel scenes unseen during training and reduce the collision rate from 2.1\% to 1.6\% in zero-shot deployment on the CARLA test set. These results show that the learned distribution is generalizable and does not overfit the training data.

---

> ### Author Response · Authors · 2024-11-25
>
> Thanks for the comments. We appreciate you find our key motivation to populate empty urban spaces in simulation “is certainly valid.” As for our experiment in real-world environments, we use all the context factors, including the ground truth goal points from the dataset. We believe the dynamic environment is already considered and addressed when obtaining these goal points, and the main focus of PedGen is to generate local body movements instead of planning the global trajectory.
>
> For your second question about the definition of the “current context,” note that our context factors include not only the static scene but also the goal point and the SMPL body shape parameter. Our main model should include all three context factors, and the model variant that only considers the scene context is only used to ablate the effectiveness of each context factor. We believe the dynamic components are already modeled in the goal context factor from another module, such as A* path planner in simulation or a motion prediction model in real-world applications, and our model focuses on using the local static scene context and the SMPL body shape to generate more plausible local body movements while reaching the goal. For example, in Fig. 8 of the supplementary, incorporating only the static scene context can help generate a sitting pose when there is a bench at the starting point and a walking upward movement when there is a slope ahead. In these cases, the static components are also critical in determining the detailed local poses other than the goal points.

---

### Official Review · Reviewer_M4G7 · 2024-10-27

**Soundness:** 4
**Presentation:** 3
**Contribution:** 3
**Rating:** 6
**Confidence:** 4

**Summary:**

The authors propose a method for learning pedestrian movements from web videos by using pre-trained predictors to generate pseudo-labels through off-the-shelf 4D human motion estimation models, despite the inherent noise in the labels. To refine these noisy labels, they introduce the PedGen model, which filters out noise and incorporates conditional inputs that may influence pedestrian behavior, thereby lifting the 2D scene into a 3D representation. The authors intend to provide open access to both the dataset and the model.

**Strengths:**

1. This paper focuses on using noisy labels to learn pedestrian motion, a novel approach with the potential to benefit various research areas.
2. The authors contribute a dataset accompanied by a label generation and filtering strategy, addressing the challenge of noise in automated labeling pipelines.
3. The results demonstrate performance improvements over baselines, and comprehensive ablation studies are conducted to validate the approach.

**Weaknesses:**

1. Although the automated labeling and filtering pipeline is essential, it is a fairly common approach, limiting the novelty of this contribution.
2. The baselines used in the comparison experiments appear weak, with only three included, potentially limiting the robustness of the results.
3. While using the goal as a conditioning factor is crucial and enhances pedestrian movement prediction, some conditions are often not visible or are difficult to capture in practical applications, such as autonomous driving. This raises concerns about the real-world applicability of the proposed setting and whether alternative solutions might address this limitation.

**Questions:**

1. How does the proposed method handle scenarios where conditional inputs are unavailable or unreliable, as might be the case in applications like autonomous driving?
2. Could the authors elaborate on why only three baselines were chosen for comparison, and whether additional baselines might provide a more comprehensive evaluation?
3. Could you clarify the specific novel aspects of the automated labeling and filtering pipeline? Additionally, is there potential for further innovation in this pipeline to enhance its originality, or were there particular design constraints that influenced its current implementation?

I am open to reconsidering my final rating if the authors address the concerns raised.

---

> ### Author Response · Authors · 2024-11-20
>
> Thank you for your thoughtful and constructive feedback. We address your questions below.
>
> **Q1**: How does the proposed method handle scenarios where conditional inputs are unavailable or unreliable, as might be the case in applications like autonomous driving?
>
> **A1**: To handle unavailable inputs, we have trained PedGen with different combinations of the context factors in Sec. 5.2. The results validate the effectiveness of adding context in each setting. To handle unreliable inputs, we have conducted additional experiments on each context factor to study its sensitivity to input noises. We add Gaussian noises with a standard deviation of 0.5 on the scene point cloud, the SMPL beta vector, and the goal position. The results are shown below:
> | Context Factor |  w/o noise  |       | |  |w/noise ($\sigma$=0.5)    |  |    |      |
> |----------------|-----------|------|------|------|------------------------|------|------|------|
> |                | mADE      | aADE | mFDE | aFDE | mADE                   | aADE | mFDE | aFDE |
> | No             | 1.13      | 4.08 | 1.61 | 7.56 | -                      | -    | -    | -    |
> | Scene          | 1.11      | 3.75 | 1.55 | 6.92 | 1.69                   | 4.02 | 2.74 | 7.41 |
> | Human          | 1.09      | 3.24 | 1.61 | 5.95 | 1.83                   | 3.50 | 3.08 | 6.22 |
> | Goal           | 0.60      | 1.09 | 0.47 | 1.00 | 1.61                   | 2.33 | 2.55 | 2.79 |
>
>
> We can see that all factors suffer from input noises, with the goal having the most degraded performance, where mADE increases by 168% and aADE increases by 114%.  The scene and the human factors have increased less in terms of the average metrics (aADE increases by 7% for scene context and increases by 8% for human context) and still performs better than the baseline without context. However, they are less robust in terms of the min metrics (mADE increases by 52% for scene context and increases by 68% for human context) as it is hard to predict the exact future movement of the dataset with noisy inputs.  We will add this experiment to our updated paper.
>
>
> **Q2**. Could the authors elaborate on why only three baselines were chosen for comparison, and whether additional baselines might provide a more comprehensive evaluation?
>
> **A2**: As context-aware pedestrian movement generation is a new task, there is no prior work that can directly support this task. Hence, we choose state-of-the-art methods from the tasks closest to our problem setting for comparison: MDM is one of the best works for action/text-conditioned human motion generation. HumanMAC is one of the most competitive methods for human motion prediction without context. TRUMANS is the state-of-the-art for indoor human-scene interaction synthesis. Please feel free to suggest other methods we are unaware of so we can compare them.
>
> **Q3**: Could you clarify the specific novel aspects of the automated labeling and filtering pipeline? Additionally, is there potential for further innovation in this pipeline to enhance its originality, or were there particular design constraints that influenced its current implementation?
>
> **A3**: We agree that our automated labeling and filtering method is common for other tasks. However, our novelty comes from **our problem setting of learning from noisy labels of web videos for diverse pedestrian movements**. This is the first time it has been attempted. In fact, existing approaches in human motion generation all assume the labels are perfect without noise. Moreover, the proposed automated label filtering is inspired by techniques addressing unsupervised anomaly detection, and we adapt them to reduce the noise level in the labels with an iterative procedure. Our experiment results in Tab.2a have shown the effectiveness of our adaptation to the new task, which reduces aADE from ***4.45*** to ***4.32*** by filtering out the anomaly labels and further reduces to ***4.08*** by adding the partial labels. For further innovation, we will identify specific parts of the motion label that have high noise, such as specific timestamps or body joints, instead of filtering out the whole motion in our future work.

---

> > ### Comment · Reviewer_M4G7 · 2024-11-25
> > **Concerns addressed**
> >
> > Thank you for the detailed and informative feedback. I appreciate the authors' efforts to address the concerns raised in my initial review.
> >
> > I recognize that this is a novel task with limited baselines available for direct comparison. The task of learning from noisy web video labels is fascinating, and the additional results provided demonstrate robustness to noisy inputs. Furthermore, the design shows adaptability in handling unavailable and varied inputs, which strengthens its practical applicability. Based on the authors’ thorough rebuttal and the new insights provided, I will increase my final rating accordingly.

---

### Official Review · Reviewer_EYrc · 2024-11-03

**Soundness:** 2
**Presentation:** 2
**Contribution:** 3
**Rating:** 8
**Confidence:** 4

**Summary:**

This paper introduces a context-aware generative model for realistic pedestrian movement prediction. It leverages a conditional diffusion framework that uses 3D point clouds to capture spatial scene context. PedGen offers a solution suitable for applications in autonomous systems, crowd simulation, and urban planning.

**Strengths:**

Context is an important factor in pedestrian trajectory prediction, which poses a challenge due to the difficulty of identifying and measuring it while predicting future paths. This paper proposes a valuable idea, supported by clear visualizations and thorough ablation studies.

**Weaknesses:**

1- I find Fig. 3 confusing. Based on the context of the paper, it appears that only one timestep is observed, and the rest are predicted. However, Fig. 3 suggests that the model is fed with the timesteps from t=1 to t=T.

2- The learnable mask *m* needs to be explained more in the paper. How is this mask learned?

**Questions:**

1- Since pedestrian path generation is done in static settings, the social attributes of pedestrians are not taken into account. Are other pedestrians considered as objects when calculating the collision measure?

2- How is the collision rate  affected by the ablations?

3- The training of masks *m* in Fig. 3 is unclear. How are these masks trained, and which part of the loss function guides this training?

3- Looking at Table 3.b, it appears that the goal has a significant effect on error reduction. However, in practice, the goal of a pedestrian is generally unknown when predicting the path. Why is the goal handled as context in this work? Shouldn’t the model predict the goal as part of the path prediction process? I am also curious to see visualizations with an ablated version where the goal is not provided as input.

4- mADE is mentioned to be measured across 50 movements. Does this mean that 50 possible paths were generated?

---

> ### Author Response · Authors · 2024-11-20
>
> Thank you for the constructive feedback. We appreciate your positive comments that this work "proposes a valuable idea, supported by clear visualizations and thorough ablation studies." We address your questions below.
>
> **Q1**: Since pedestrian path generation is done in static settings, the social attributes of pedestrians are not taken into account. Are other pedestrians considered as objects when calculating the collision measure?
>
> **A1**: As mentioned in the global response, our model serves as the first method for the new task of pedestrian movement generation for synthesizing realistic pedestrian movements in simulation, and we use the static scene context to better support populating empty environments. To evaluate generation in simulation, we only experimented on static scenes without dynamic objects in the CARLA simulator to measure the collision rate. The experiment shows that our model already implicitly learns the social attributes of pedestrians. As shown in Tab.1, PedGen improves mADE from ***1.31*** to ***1.13*** on CityWalkers and from ***3.03*** to ***2.90*** on Waymo in real-world scenarios by conditioning on the scene context, which contains the other pedestrians' point cloud. We could further iteratively generate future movements based on the context at the latest time step so the model can adaptively update its predicted movements according to the behaviors of other pedestrians.
>
>
> **Q2**: How is the collision rate affected by the ablations?
>
> **A2**: An additional ablation experiment in the CARLA test set shows that adding context factors reduces the collision rate and improves the physical plausibility. See the table below:
> | Context Factor | Collision Rate %  | Foot Floating Rate %|
> |----------------|-----|-----|
> | No             | 2.1 | 5.2 |
> | Scene          | 1.6 | 2.6 |
> | Human          | 1.9 | 0.7 |
> | Scene+Human    | 1.5 | 0.3 |
> | Goal           | 0.0 | 0.0 |
>
>
> The scene context can contribute to both a lower collision rate of ***1.6%*** and a lower foot floating rate of ***2.6%***, while the human context is more useful in reducing the foot floating rate to only ***0.7%***. Adding both the scene and the human context can further improve the physical plausibility of the generated movements.  Using the goal context is the most crucial factor and can reduce the failure rate to 0. We will add this experiment to our updated paper.
>
> **Q3**: The training of masks *m* in Fig. 3 is unclear. How are these masks trained, and which part of the loss function guides this training?
>
> **A3**:  To train the partial labels with masking embeddings $m$, we first define a label mask $\boldsymbol{M}$, where $\boldsymbol{M}_t==1$ indicates the label at timestep $t$ is missing. Then we add the mask embedding to the original noisy sample as $\boldsymbol{x}^k = \boldsymbol{x}^k(1-\boldsymbol{M}) + \boldsymbol{m}\cdot \boldsymbol{M}$ to replace the missing timesteps with the mask embedding $\boldsymbol{m}$ and feed to the network to output the denoised prediction $\hat{\boldsymbol{x}}$ similar to Sec.4.2.  We then update the loss with the masked predictions and masked ground truth as $L = L(\boldsymbol{x}(\boldsymbol{M}), \hat{\boldsymbol{x}}(\boldsymbol{M}))$ so it only operates on the labels at the available timesteps. All losses will guide the training with the masked outputs. We will update the corresponding section in our paper to make the training of masks $\boldsymbol{m}$ clearer.
>
> **Q4**: Looking at Table 3.b, it appears that the goal has a significant effect on error reduction. However, in practice, the goal of a pedestrian is generally unknown when predicting the path. Why is the goal handled as context in this work? Shouldn’t the model predict the goal as part of the path prediction process? I am also curious to see visualizations with an ablated version where the goal is not provided as input.
>
> **A4**: As mentioned in the global response, our work focuses on generating realistic and detailed pedestrian local movements as SMPL body meshes and synthesizing pedestrian animations in simulation. Therefore, we assume the global path is given by a path planner. We will add visualizations to compare the model generation without the goal context to further demonstrate the effectiveness of the other context factors.
>
> **Q5**: mADE is mentioned to be measured across 50 movements. Does this mean that 50 possible paths were generated?
>
> **A5**: Yes, we generate 50 possible movements, and mADE measures the minimum ADE (average displacement error) among the 50 movements compared to the ground truth.

---

> > ### Comment · Reviewer_EYrc · 2024-11-25
> >
> > Thank you for responding to questions.
> >
> > -I see Figure 8 in Appendix with two rows (No context and Scene only). Which of these figures are the result of Goal ablation?
> >
> > -Why is the collision rate (and its values after ablations) only calculated in CARLA and not in the two other datasets?

---

> ### Author Response · Authors · 2024-11-25
>
> Thanks for the comments. We will address your further questions.
>
> **Q1.** I see Figure 8 in Appendix with two rows (No context and Scene only). Which of these figures are the result of Goal ablation?
>
> **A1.** Sorry for the confusion. We have updated Fig. 8 to add visualizations with both the scene and the goal context for a better comparison. The figure shows that the model using both the scene and the goal context can reach the goal precisely, while the model can still generate plausible pedestrian movements when only using the scene context  (especially for the sitting and standing poses in Fig. 8b and Fig. 8d). The model would perform poorly when no context factor is provided. The qualitative results further demonstrate the effectiveness of the scene context when the goal is not provided as input.
>
> **Q2.** Why is the collision rate (and its values after ablations) only calculated in CARLA and not in the two other datasets?
>
> **A2.** We only calculate the collision rate in CARLA as it is a simulator that has ground truth scene geometry and a collision checker. On the contrary, it is challenging to compute the collision rate in real-world datasets due to a lack of ground truth labels for the scene geometry, like the ground height and meshes of the obstacles. Therefore, we only evaluate the ADEs and MDEs on these datasets by comparing with the ground truth pedestrian movement.

---

> > ### Comment · Reviewer_EYrc · 2024-11-26
> >
> > Thank you for updating Figure 8 in the appendix.
> >
> > - Could you please include the first timestep movement (the observed timestep) with a slightly different color? It seems like figure 8 only includes the generated movements. I am trying to understand why there is a significant shift in the direction and pose between each of these generated movements.

---

> > > ### Author Response · Authors · 2024-11-26
> > >
> > > Thanks for the suggestion. We have updated Figure 8 in the appendix, and now the first timestep pose is white.  Note that the initial pose in the first timestep is also generated instead of observed, as our model only requires the initial starting point of the movement. Therefore, the generated initial pose is also influenced by the input context factors, so there is a significant shift in the initial pose between using the context and not using the context. Generating the initial pose makes our model flexible and facilitates populating an empty simulation environment.

---

> > > > ### Author Response · Authors · 2024-11-27
> > > >
> > > > Dear Reviewer EYrc,
> > > >
> > > > Thanks again for your insightful comments! We believe we have addressed your comments and are keen to participate in the discussion with you.
> > > >
> > > > We would appreciate it if you would let us know whether our responses sufficiently address your questions and whether you need to see additional visualizations, as the last day to update the materials is Nov. 27th. Thank you once again!
> > > >
> > > > Best regards,
> > > > Authors of Submission ID 1930

---

### Author Response · Authors · 2024-11-20
**Global Response**

We appreciate all reviewers for the insightful and helpful feedback. We would like to first emphasize our problem setting and the motivation behind it. Our paper aims to address the new task of pedestrian movement generation and **apply it to the simulation environment to synthesize realistic pedestrian animations**. Unlike existing works that tackle long-term pedestrian trajectory prediction, pedestrian movement generation focuses more on learning realistic body movements for the SMPL meshes from real-world videos. Therefore, in Sec.2, we have defined our task as “***pedestrians continuously make short-term movement decisions on their route to respond to their immediate environment***” in reference to the literature on pedestrian behavior analysis (Feng et al., 2021). To support the application of PedGen to a simulator, we have made two key assumptions to our problem setting.

First, we assume the global trajectory can be given by another module so we can focus on generating realistic pedestrian body movements. For example, we can use a path planner like A* to generate a plausible path in the simulator. Our model can run sequentially to these modules and use their output goal points as the goal context. Its key novelty is that it can transfer natural and diverse real-world movements to simulation with more degrees of freedom instead of relying on fixed animations in existing simulators (Shan et al., 2023).  Second, we only consider the static scene context at the current frame to support populating an empty environment in a simulator with no history information.  In our experiments in Tab. 2b, we have shown the effectiveness and importance of incorporating each context factor compared to using no context. Despite PedGen only having one static scene context, the result in Tab. 2b already shows that it can provide PedGen ***6.7%*** and ***8.5%*** improvement in aADE and aFDE compared to the one without it. With a simple modification of stacking history point clouds, our scene context encoder can also encode multi-frame context information.

---

> ### Author Response · Authors · 2024-11-24
> **Paper Update**
>
> Thanks again for all reviewer's suggestions on our work.
> We have updated our submission. Here's the summary of the modifications to our paper to address some of the questions.
> 1.  We have added more formulas to the training with the mask embedding $\boldsymbol{m}$ to make it clearer in Sec. 4.3, as suggested by Reviewer EYrc Q3.
> 2. We have conducted additional experiments to show how each context factor would affect the collision rate and the foot floating rate in the CARLA simulator in Tab. 2b, as asked by Reviewer EYrc Q2.
> 3. We have added additional visualizations to better show the utility of the scene context compared to not using the context in Fig. 8 in the appendix, as asked by Reviewer EYrc Q4 and Reviewer yQtW Q4.

---

### Meta-Review · Area_Chair_Tqr9 · 2024-12-17

**Metareview:**

This paper introduces a context-aware generative model for realistic pedestrian movement prediction. This paper proposes a valuable idea, supported by clear visualizations and thorough ablation studies. The results demonstrate performance improvements over baselines, and comprehensive ablation studies are conducted to validate the approach. The major concerns of reviewers include implementation details, methodological reasonableness, unclear task definitions, and clarification of contributions. The author's response has addressed most of these issues. So the final vote is acceptance.

**Additional Comments On Reviewer Discussion:**

The authors improved the readability of the paper during the rebuttal, including the addition of method descriptions, additional experiments, and visualized results.

---

### Decision · Program_Chairs · 2025-01-22

Accept (Poster)